# Untargeted Metabolomics Combined with Sensory Analysis to Evaluate the Chemical Changes in Coppa Piacentina PDO during Different Ripening Times

**DOI:** 10.3390/molecules28052223

**Published:** 2023-02-27

**Authors:** Gabriele Rocchetti, Alessandra Scansani, Giulia Leni, Samantha Sigolo, Terenzio Bertuzzi, Aldo Prandini

**Affiliations:** 1Department of Animal Science, Food and Nutrition, Università Cattolica del Sacro Cuore, Via Emilia Parmense 84, 29122 Piacenza, Italy; 2Consorzio Tutela Salumi DOP Piacentini, Via Tirotti 11, 29122 Piacenza, Italy

**Keywords:** meat quality, metabolomics, sensory quality, lipid oxidation, protein oxidation

## Abstract

Ripening time is known to drive the chemical and sensory profiles of dry meat products, thus potentially affecting the final quality of the product. Starting from these background conditions, the aim of this work was to shed light, for the first time, on the chemical modifications of a typical Italian PDO meat product—namely, Coppa Piacentina—during ripening, to find correlations between its sensory quality and the biomarker compounds related to the progress of ripening. The ripening time (from 60 to 240 days) was found to deeply modify the chemical composition of this typical meat product, providing potential biomarkers of both oxidative reactions and sensory attributes. The chemical analyses revealed that there is typically a significant decrease in the moisture content during ripening, likely due to increased dehydration. In addition, the fatty acid profile showed that the distribution of polyunsaturated fatty acids significantly (*p* < 0.05) decreased during ripening, because of their high susceptibility to oxidation and conversion to intermediate and secondary molecules. An untargeted metabolomics approach, coupled with unsupervised and supervised multivariate statistics, highlighted a significant impact (prediction scores > 1) of lipid oxidation during ripening time, with some metabolites (such as γ -glutamyl-peptides, hydroperoxy-fatty acids, and glutathione) being particularly discriminant in predicting the changes observed. The discriminant metabolites were coherent with the progressive increase of peroxide values determined during the entire ripening period. Finally, the sensory analysis outlined that the highest degree of ripening provided greater color intensity of the lean part, slice firmness, and chewing consistency, with glutathione and γ-glutamyl-glutamic acid establishing the highest number of significant correlations with the sensory attributes evaluated. Taken together, this work highlights the importance and validity of untargeted metabolomics coupled with sensory analysis to investigate the comprehensive chemical and sensory changes to dry meat during ripening.

## 1. Introduction 

Coppa Piacentina is a typical Italian product characterized by a protected designation of origin (PDO) and registered with the publication of Regulation (EC) n. 1263/96. Pigs born, reared and slaughtered in Lombardy and Emilia Romagna at 160 kg, +10% live weight (LW), and aged at least nine months comply with the requirements already established at the national level for Parma and San Daniele PDO hams. Furthermore, the pigs must present with the characteristics of the Italian heavy pig, defined in accordance with Regulation (EC) n. 1237/07 concerning the commercial classification of pig carcasses. The product, Coppa Piacentina PDO, derives from the cervical muscle of the pig and, fresh, must have a weight of no less than 2.5 kg. The production area of Coppa Piacentina PDO includes the entire territory of the province of Piacenza, limited to areas at an altitude of less than 900 m above sea level, due to the particular climatic conditions. The technological process begins with dry salting with a mixture of salts and natural aromas, and after resting in the refrigerator they are coated with a diaphragm porcine parietal. This is followed by binding, drying and aging for a minimum of six months. When released for consumption, it has a cylindrical shape on the outside, slightly thinner at the end, with a compact, non-elastic consistency. When cut, the slice is compact and homogeneous, red in color interspersed with pinkish white on the marbled parts. The weight must not be less than 1.5 kg. The scent is sweet and it has a sweet and delicate taste which is refined with the progress of maturation. Coppa Piacentina is also an important Italian PDO product from a marketing point of view; in 2021, 790 tons were marketed, which equates to 490,000 certified pieces. 

Where the study of meat products is concerned, it is important to mention that several biochemical changes are known to occur during the ripening of dry meat [1]. Drying is one of the most crucial food preservation techniques, which involves reducing the moisture levels of products to the necessary moisture content in order to avoid spoiling and retain quality. It is a process that simultaneously transfers heat and moisture, which causes changes in the product that is being dried [2]. Meat drying is the removal of water content from the meat surface in order to obtain an ideal condition for dried meat product preservation. Drying conditions should be thoroughly considered as they will affect the quality of the meats after going through processes which may alter the quality or condition of the meats that people desire to preserve [2]. In particular, temperature, drying period, relative humidity, and water content are crucial parameters in the drying conditions that will affect the quality of the dried meats. In choosing the proper drying process, optimizing the drying temperature is an essential element influencing the quality of dried meat products. Within this context, several advanced drying techniques have been developed and studied extensively, such as ultrasound-assisted vacuum drying and freeze drying, which aim to improve the quality and sensory aspects of the dried meat under consideration. The changes that occur during the ripening of dry meat can modify the flavor and odor of the final product [3]. Microbial growth combined with the activity of endogenous enzymes can enhance the development of a number of aromatic and sapid compounds in the meat matrix [4]. Lipid auto-oxidation reactions are also an important source of these compounds, and it is not well understood which of these processes is hierarchically more impactful during ripening time [5]. In the last few years, a huge amount of research has been focused on the hydrolysis of triglycerides into free fatty acids, diglycerides, and monoglycerides during ripening, together with the progressive increase in the amounts of different carbonyl oxidation products [6]. Carbonyl compounds play a significant role related to flavor because, in general, these compounds have very low perception thresholds (i.e., in the ppm/ppb range). Similarly, the breakdown of proteins to yield peptides and amino acids is highly important during ripening time, peptides and amino acids being preferential substrates of several microbial and chemical reactions that generate many flavor compounds [1,4]. 

Considering the several reactions occurring during ripening, it is important to take an analytical approach able to comprehensively evaluate a wide spectrum of chemical classes and subclasses. Recently, metabolomics based on chromatography, coupled with high-resolution mass spectrometry, has been proposed to investigate the qualitative traits of meat and meat products [7,8,9]. This approach allows the putative and simultaneous identification of a wide range of metabolites and small peptides, as related to different factors (such as processing, ripening, and shelf-life conditions) [10,11]. Metabolomics is a relatively new component in systems biology that focuses on the high-throughput characterization of small molecular metabolites in biological systems [12]. It is widely used in several scientific fields, particularly food science. Due to its excellent detection and prediction capacities, metabolomics is well suited to the analysis of complex matrices. With the development of a new generation of mass analyzers, the resolution quality has been improved significantly. Currently, mass spectrometry (MS), depending on its mass analyzers, can be divided into Magnetic Sector MS, Quadrupole MS, Ion Trap MS, time of flight MS (TOF-MS), Fourier transform Ion cyclotron resonance MS (FTICR-MS), and Orbitrap MS, which emerged in recent years. Among these MS analyzers, TOF-MS, FTICR-MS, and Orbitrap MS are High Resolution Mass Spectrometer (HRMS) with resolution ≥10,000, which can be used for accurate qualitative and non-directional unknown screening in food samples [12]. In addition, it plays an unprecedented role in the rapid detection of food composition because of its fast scanning time and positive/negative ion capability, as well as its ability to provide exact mass and possible elemental composition of precursor and fragment ions, aiding in the identification of unknown compounds [12]. 

Starting from these background conditions, the aim of this work was to comprehensively investigate, for the first time, the biochemical changes occurring during the ripening time of a typical Italian PDO product, namely Coppa Piacentina. The novelty of this study lies in the utilization of an untargeted metabolomics-based approach, combined with classical chemical analysis and a sensory evaluation carried out by trained panelists. In particular, this approach was used to find potential correlations between the chemical signatures and the sensorial profile of the product, thus proposing potential biomarkers of the meat product.

## 2. Results and Discussion

### 2.1. Proximate Composition of Coppa Piacentina PDO Samples

Table 1 reports the effect of ripening on the composition (i.e., moisture, proteins, lipids, and total peroxides) of Coppa Piacentina PDO samples. Overall, a significant decrease in moisture content was detected during ripening, with values ranging from 62.3% at T0 to 33.8% after 240 days of ripening, likely due to the increased dehydration that occurred during aging. These findings agree with the general rule that moisture content is inversely related to ripening time in cured meat products, as also previously reported by Andres et al. [13]. Conversely, protein and lipid content significantly increased during the ripening time, due to the decreased moisture content which leads to an increase in dry-matter content in Coppa Piacentina PDO samples. However, no significant differences were determined when the data were expressed on dry matter. 

### 2.2. Modification of Fatty Acid Profile and Total Peroxides during Ripening

The progression of lipid oxidation following the production and ripening of Coppa Piacentina PDO has been evaluated by determining the peroxide value, being a key marker of primary lipid oxidation. As shown in Table 1, total peroxide values significantly increased with the increasing ripening time, ranging from 0.6 mEq. O_2_/kg lipid for the raw sample (T0) to 26.0 mEq. O_2_/kg lipid after 240 days of ripening. These data agree with the values reported by Cava et al. [14], who analyzed the peroxide value of pork muscle before and after a long ripening period (a total of 700 days), obtaining values which ranged from 1.1 to 20.9 mEq. O_2_/kg lipid. Indeed, during ripening, the combination of high temperatures, long drying times and high salt content enhance and promote lipid oxidation [15]. Besides the oxidation phenomena, lipolysis and changes in fatty acid composition have also previously been reported in dry-cured pork products [16]. For this reason, in the present work and for the first time, the changes to the fatty acid profile characterizing Coppa Piacentina PDO were investigated across 240 days of ripening, and the results are reported below (Table 2).

As far as the fatty acid profile is concerned, MUFA and SFA were the main fatty acid groups found in the lipid fraction, representing more than the 85% of the profile, as also reported for other dry-cured pork products [5,17]. In relation to the individual fatty acid profile, oleic acid (C18:1n-9) was the most predominant fatty acid detected in all samples of Coppa Piacentina PDO, representing on average 40% of the whole lipid fraction. Its distribution significantly changed (*p* < 0.05) during ripening, ranging from 37.20% in the raw sample (T0) to 43.10% after 240 days. Besides oleic acids, palmitic acid (C16:0), stearic acid (C18:0) and linoleic acid (C18:2n-6) were detected in high concentrations in all samples, together accounting on average for 51% of the whole fatty acid profile. Therefore, these results agree with the ones reported by Muzolf-Panek et al. [18], who determined the same order of fatty acid abundance in raw pork neck samples. In the current work, for the first time, the change in the fatty acid composition of Coppa Piacentina PDO during the ripening period was studied. Linoleic acid (C18:2n-6), linolenic acid (C18:3n-3), eicosadienoic acid (C20:2), eicosatrienoic acids (C20:3n-6 and C20:3n-3), arachidonic acid (C20:4n-6) and docosapentaenoic acid (C22:5n-3), all belonging to the PUFA group, significantly decreased (*p* < 0.05) during ripening. A similar trend was also seen for heptadecanoic acid (C17:0), the only SFA which was significantly affected (*p* < 0.05) by ripening and whose distribution decreased by about 10% from T0 to T240. On the other hand, the percentage of oleic acid (C18:1n-9) and octadecenoic acid (C18:1cis11) significantly increased (*p* < 0.05) over the ripening time, reaching the highest values after 240 days at 43.10% and 2.95%, respectively. Ripening did not significantly (*p* > 0.05) influence the distribution of fatty acids among the SFA and UFA groups, recording average values of 42.71 and 57.29%, respectively. However, the proportion of MUFA and PUFA significantly changed (*p* < 0.05) over 240 days of ripening. In particular, PUFA significantly decreased during the ripening, representing 14% of the total fatty acids in the raw meat samples, while making up only 9% after 240 days of ripening. Moreover, as a consequence, a significant increase in MUFA was detected from the raw meat (T0 sample) to the end of ripening (T240 sample), where the proportion ranged from 42% to 49% of the total fatty acids. These results agreed with the ones reported by Seong et al. [19] in dry-cured pork loins and by Cava et al. [20] in Iberian hams, where a significant decrease in PUFA distribution was detected at the end of both ripening periods. Also, Larrea et al. [21] determined that in Teruel dry-cured ham there was a significant drop in the percentage of PUFA after 6-8 months of ripening, with values which decreased from 16% in the raw meat to 12% in the final product. The PUFA decrease is related to their high susceptibility to oxidation and their related conversion to intermediate and secondary molecules [5]. In particular, PUFA react with molecular oxygen via a free radical mechanism, producing hydroperoxides, which are considered to be the first oxidation products. As a matter of fact, as PUFA decreased over the ripening time, the total peroxides significantly increased during ripening. However, only a moderate correlation (R = 0.430) was determined between PUFA and peroxide values, underlining that the instability of hydroperoxides could have led to their rapid conversion to other secondary compounds (e.g., hydrocarbons, aldehydes, ketones, alcohols, esters and acids) [5]. UFA, especially PUFA, are well known as healthy fats that have important physiological functions in maintaining a healthy life, such as helping brain development, offering protection against cardiovascular and neurological diseases, and possessing antiapoptotic and anti-inflammatory properties [22]. In the present study, however, the extension of ripening time up to 240 days caused a decrease in PUFA levels. Therefore, solutions to prevent this loss due to oxidation during the ripening process should be considered in order to assure the nutritional value of this product. 

### 2.3. Marker Compounds of the Ripening Time by Untargeted Metabolomics

In this work, untargeted metabolomics based on ultra-high-performance liquid chromatography, coupled with high-resolution mass spectrometry (UHPLC-HRMS), was used to provide a comprehensive screening of the chemical composition of Coppa Piacentina PDO samples during different ripening times. Overall, the approach used allowed the identification of 1052 compounds according to a level 2 confidence in annotation (i.e., putative identification + structural confirmation of the Top N ions). All the compounds annotated are reported in the Appendix A) including their name, total identification (ID) score, reference m/z, chemical formula, INCHIKEY code, MS1 isotopic spectrum, and MS/MS spectrum. By exploiting the information gained from quality control (QC) samples (i.e., annotated by data dependent MS/MS approach) we structurally confirmed 234 metabolites against the comprehensive database FoodDB (Appendix A). Overall, consistent with the nature of the matrix analyzed (i.e., meat samples), we found a great abundance of amino acids and peptides (showing the highest significant enrichment ratio; Appendix A), followed by several lipid subclasses, such as glycerophospholipids, fatty acids and conjugates, fatty esters, prenol lipids, fatty esters and amides, diradylglycerols, purines, and pyrimidines (Figure 1).

Among those compounds confirmed by data-dependent MS/MS we found 10 amino acids, namely alanine, arginine, cysteine glutamic acid, histidine, isoleucine, phenylalanine, proline, tryptophan, and tyrosine (Appendix A). Furthermore, the analysis allowed the structural confirmation of creatine and its metabolic by-product creatinine, two key metabolites facilitating a stable and rapid supply of energy to the muscle. Interestingly, other structurally confirmed metabolites were tyramine, a biogenic amine produced from the decarboxylation of tyrosine, and spermine, a biogenic amine derived from putrescine [23]. Regarding fatty acid composition, we provided a structural confirmation of alpha-linolenic acid and dihomolinolenic acid (omega-3 fatty acids), together with 9,10-dihydroxy-12Z-octadecenoate (9,10-DHOME; a derivative of linoleic acid diol). Additionally, some aldehydic compounds were annotated and confirmed, such as 2,6-dimethyl-5-heptenal and 9-octadecenal. 

As far as the impact of ripening time is concerned (i.e., a total period of 240 days), we carried out a multivariate statistical approach based on both unsupervised and supervised methods, namely principal component analysis (PCA) and orthogonal projections to latent structures discriminant analysis (OPLS-DA), respectively. Regarding the unsupervised PCA approach, the score plot (Figure 2) revealed a clear chemical difference when considering T0 vs ripened Coppa Piacentina PDO samples (T60, T90, T180, and T240). Overall, the PCA allowed us to better understand the variability explained by the annotated metabolites; in this regard, the two-principal components (PC1 and PC2) were found to explain 46% of the total variability. Interestingly, the PCA score plot also showed the different chemical behavior of the samples belonging to intermediate ripening periods (i.e., T60 and T90) when compared with long-ripened samples (T180 and T240). 

The supervised OPLS-DA statistics were then used to shed light onto the contribution of each main group of discriminant metabolites. Figure 3 shows the cross-validated OPLS-DA score plot built to compare the metabolomic profiles of the different Coppa Piacentina PDO samples during the ripening period. The model highlighted a clear separation between the different samples at each time point considered, thus supporting the results revealed by unsupervised statistics. In this regard, the introduction of an orthogonal latent vector allowed us to better discern the chemical differences when considering samples belonging to T180 and T240 groups (Figure 3). Therefore, both unsupervised and supervised multivariate statistics confirm the potential of untargeted metabolomics to discern the impact of the ripening period on the chemical modifications of this typical Italian PDO product. The OPLS-DA model had acceptable cross-validation parameters, being R^2^Y (cum) = 0.976, and Q^2^(cum) = 0.837, and the permutation test cross-validation (N = 200; Appendix A) excluded over-fitting. Also, the analysis of variance (ANOVA) on cross-validated residuals showed a significant *p*-value (*p* < 0.05). 

The variable’s importance in projection (VIP) selection method was then used to extrapolate the most discriminant metabolites during the ripening process, i.e. those compounds driving the main differences observed. Overall, 548 discriminant metabolites showed a VIP score > 1 (i.e., high prediction score), thus driving the sample separation observed in Figure 3. Overall, amino acids, peptides, and analogues were the most represented group, followed by fatty acids and glycerolipids. A detailed list containing the discriminant compounds together with their VIP scores, organized in chemical classes, is provided in Appendix A. Interestingly, five compounds presented the best discriminant potential during ripening, namely arginine (belonging to amino acids; VIP score = 1.77), dodecanedioylcarnitine (belonging to esters of fatty acids; VIP score = 1.74), 3-dehydroxycarnitine (belonging to fatty acids; VIP score = 1.62), PG(16:0/22:4(7Z,10Z,13Z,16Z)) (belonging to glycerophospholipids; VIP score = 1.59), and asparaginyl-phenylalanine (belonging to peptides; VIP score = 1.57).

After that, considering the great impact of oxidation processes during ripening time (240 days), as revealed by the analysis of total peroxides (Table 1), we extrapolated the most discriminant compounds dealing with lipid and protein oxidation, which are reported in Table 3. 

Accordingly, 16 compounds were found to possess a high discriminant potential when considering their potential role in oxidation events during the ripening process, including amino acids, peptides, aldehydic compounds, hydroperoxy-derivatives of fatty acids, and polyamines. As far as the presence of polyamines is concerned, spermidine and tyramine are a ubiquitous group of compounds involved in many biological processes, including binding to nucleic acids, stabilizing membranes, and stimulating several enzymes. Putrescine is converted to spermidine by spermidine-synthase through the addition of a propylamine group derived from the decarboxylation of S-adenosyl-methionine. Besides, as previously stated, tyramine is derived from the decarboxylation of the aminoacid tyrosine [24]. In this work, both spermidine and tyramine showed high VIP scores (>1), thus confirming their importance during ripening (Figure 3). However, by inspecting the Log_2_ Fold-Change (FC) values, we found that these compounds were almost stable during ripening (Table 3), although up-accumulated for each comparison against T0 (fresh meat), likely indicating potential metabolic activity by the endogenous microbiota. The interconversion of polyamines is a cyclic process that controls their turnover and regulates intracellular homeostasis. These compounds could be due to both the matrix (meat) and the metabolic activity of endogenous microbiota characterizing this naturally fermented, ripened, and dry meat product [25].

Few studies are available in the scientific literature regarding the chemical and microbiological profile of Coppa Piacentina. A previous work by Busconi et al. [26] reported for the first time the bacterial ecology of Coppa Piacentina PDO samples at the end of ripening. In particular, the authors showed that at T0 the most abundant bacterial genera were Lactobacillus and Staphylococcus, with *L. sakei*, *L. curvatus*, *S. xylosus*, *S. equorum*, and *Tetragenococcus halophilus* being the most represented species. Therefore, the detection of both aromatic and aliphatic polyamines (Table 3), could be potentially due to lactic acid bacteria, which are considered the main biogenic amine producers in fermented foods. In particular, besides Enterococci (constituents of the natural microbiota of raw meat and fermented meat products, with *E. faecalis* and *E. faecium* being the predominant species), the main tyramine producers in fermented meat products are strains belonging to *L. curvatus* species, which together with *L. sakei* is the predominant Lactobacillus species [27]. Therefore, the increase of tyramine during ripening is coherent with the natural microbiota reported in the literature for Coppa Piacentina PDO.

As far as the presence of *S. xylosus* is concerned, it is widely known that this strain is well adapted to the meat matrix and it finds wide utilization as a starter culture in fermented meat products [26,28]. In fermented meat, Staphylococci are essential for color formation and rancidity prevention. Methyl ketones from β-oxidation of fatty acids and methyl-branched volatiles from amino acids (such as valine, leucine and isoleucine) have been correlated with the typical dry sausage flavor; these develop very slowly, being subjected to the α-ketoglutarate availability. This latter is strictly connected with the glutamate homeostasis, which is really important in dry fermented meat products. In this regard, glutamate dehydrogenase activity provides α-ketoglutarate, an important intermediate required for amino acid transamination, which initiates the conversion of amino acids to aromatic compounds. Glutamate could be synthesized using the glutamate synthase from α-ketoglutarate, an intermediate of TCA cycle, and glutamine. Glutamate is the major amino group donor for all nitrogen-containing compounds, as a link between nitrogen and carbon metabolism. Therefore, the glutamate-synthesizing and -degrading reactions must be tightly controlled to maintain its homeostasis. In a previous work by Vermassen et al. [28] regarding the adaptation of *S. xylosus* in a salted meat model, the authors found an overall up-regulation of genes involved in glutamate homeostasis, thus confirming the important role of this amino acid in the meat matrix under potential osmotic stress. This aspect has also been considered to be consistent with the major flavor richness of naturally fermented long-ripened dry sausages. Accordingly, the contribution of the staphylococcal population to flavor development has also been documented in dry-cured ham [26]. Interestingly, the untargeted metabolomics-profiling also outlined the importance of glutamate as related to glutathione and γ-glutamyl peptides, namely γ -glutamyl-L-putrescine, γ -glutamyl-glutamic acid, and γ -L-glutamyl- γ -L-glutamyl-L-methionine (Table 3). γ-Glutamyl peptides are small molecular peptides that greatly influence the perception of kokumi taste, a sensation that promotes the palatability of food, and it can be specifically defined as a sensation that imparts “continuity”, “mouthfulness”, “thickness”, and “richness” [29]. As small molecular peptides, γ-glutamyl peptides can be obtained via dehydration condensation of the γ-carboxyl group of glutamic acid or other γ-glutamyl compounds (e.g., glutamine and glutathione) with the α-amino group of amino acids or peptides [4]. The kokumi threshold concentrations have been identified for some peptides; according to Wang et al. [29], γ -glutamyl-glutamic acid has a kokumi threshold concentration of 17.5 μmol/kg, while γ -L-glutamyl-γ-L-glutamyl-L-methionine is reported to have a higher threshold concentration, being on average 650 μmol/L. According to the literature [30], pathways for the generation of some γ-glutamyl peptides have been presented and enzymes directly related to these pathways have been investigated. However, the specific formation mechanism of these peptides in natural and processed foodstuffs remains unclear. In addition, to further improve food processing and enhance the desirable characteristics of food, a better understanding is required of the interrelation between these peptides and the taste quality of food. As recently reviewed by Wang et al. [29], the application of a metabolomics approach to explore compounds related to kokumi intensity will be helpful in illustrating the mechanism of kokumi γ-glutamyl peptide formation. Therefore, our preliminary study supports the need to perform future evaluations on the importance of these peptides and their sensory perception.

Finally, untargeted metabolomics also provided new insights into the lipid oxidation phenomena of Coppa Piacentina PDO samples, already highlighted by looking at the number of total peroxides detected (Table 1). Overall, the oxidation of meat components in ripened products is a challenging process from a biochemical point of view. On the one side, it is essential since the enzymatic and non-enzymatic lipid oxidation creates flavor and aroma compounds characteristic for ripening products; on the other side, lipid oxidation can generate undesired compounds. Under our experimental conditions, we detected a significant and marked increase in the hydroperoxy- and oxo-derivatives of unsaturated fatty acids (i.e., linoleic and linolenic fatty acid derivatives), and this was particularly evident after 240 days of ripening (Table 3). Accordingly, the compounds (9S,10E,12Z)-9-hydroperoxy-10,12-octadecadienoate, (9Z,11E,13E,15Z)-4-oxo-9,11,13,15-octadecatetraenoic acid, and (9Z,11E,14Z)-(13S)-hydroperoxyoctadeca-(9,11,14)-trienoate recorded Log_2_FC values of 5.19, 6.50, and 3.07, respectively. Therefore, in this work, the marker compounds revealed by untargeted metabolomics were effective in evaluating the peroxidation of lipids. It is known that the unsaturated fatty acids characterizing meat can react with molecular oxygen via a free radical mechanism to produce so-called hydroperoxides [9]. Indeed, these latter are considered the first oxidation products of the oxidation pathway, being practically odorless and providing a scarce contribution to the final aroma. In addition, the strong instability of hydroperoxides and their rapid decomposition can generate a high number of secondary reaction products, including hydrocarbons, aldehydes, ketones, alcohols, esters, and acids, which cause the development of off-flavors and off-odors in meat. In this work, we found three aldehydes significantly related to the ripening process, namely 8-heptadecenal, 9-octadecenal, and 9-tetradecenal (Table 3), again recording maximum increases after 240 days of ripening. The accumulation of 9-octadecenal is coherent with the abundance of MUFA (mainly C18:1n9c) outlined in Table 2. Aldehydes are considered the most important breakdown products and the largest contributors to volatile flavors in meat. They usually show low odor thresholds and a significant distribution in meat samples undergoing peroxidation reactions [5]. On the one side, aldehydes are related to the aromatic bouquet of meat products, while on the other side they can react with proteins thus potentially affecting their organoleptic properties. Finally, as far as the putative annotation of glutathione is concerned, it is well known that the glutathione system is the major endogenous antioxidant machinery that protects animal cells from oxidative damage. Our findings revealed a significant and progressive reduction in glutathione as ripening time progressed, recording a Log_2_FC value equal to −8.14 at 240 days of ripening (Table 3). The extremely significant and negative Log_2_FC values measured for glutathione demonstrate a marked redox impairment of Coppa Piacentina PDO samples during ripening, likely counteracting the lipid peroxidation and protein oxidation and being involved in the exclusive aroma development of the product [4,5,31].

### 2.4. Effect of Coppa Piacentina PDO Ripening Period on Sensory Characteristics

During the ripening of dry-cured hams, intensive biochemical reactions occur, leading to a significant change in the composition and, consequently, in the sensorial properties of the product. The sensory properties of dry-cured meat products are related to intramuscular lipid composition and to the extent of proteolysis, lipolysis and the lipid oxidation during processing [32]. In order to investigate the effect of ripening on the sensory characteristics of Coppa Piacentina PDO, a panel test was performed and the mean scores for sensory characteristics are reported in Figure 4.

As far as the sensory test is concerned, 17 sensory attributes were assessed by the panelists. The Coppa Piacentina PDO sample collected after 60 days of ripening was considered only for visual and olfactory aspects, while it was not considered for taste aspects. For this reason, Figure 3 provides a mean score value of “0” for the sweet, salty, chewing consistency, global positive and negative aroma, aromatic persistency, and taste pleasantness characteristics of the T60 sample. Interestingly, samples with the highest degree of ripening (i.e., 240 and 180 days) were characterized by greater color intensity of the lean part, greater slice firmness and higher chewing consistency. In particular, the 240-day-ripened samples were perceived to be the most intense in terms of the color of the lean part, while samples with a lower ripening degree (60 days) showed a lower color saturation, although a more uniform color could be detected. These trends agree with those phenomena occurring during ripening, i.e. color tending to darken, and greater surface dehydration causing a more intense color on the outside part, with a corresponding decrease in the color uniformity of the slice. Similarly, the highest slice firmness score detected for T180 and T240 samples was ascribable again to the dehydration of meat which occurs during ripening, perfectly in agreement with the results regarding moisture content reported in Table 1. Regarding the Coppa Piacentina PDO samples undergoing a shorter ripening period (i.e., 60 and 90 days), a more intense aroma of fresh meat was perceived, which on the contrary was not detected after 180 days of ripening. Additionally, the panelists identified in the long-ripened sample (T240) a slight rancid smell, which was not detected in the T60 and T90 samples. Therefore, the sensory analysis results confirmed what was previously highlighted by the different chemical analyses, i.e. the hierarchically higher role of the oxidative phenomena driving the sensory profile of the product. In this regard, it is known that with the progress of aging the oxidative phenomena intensify, sometimes resulting in the appearance of rancid odors, thus perfectly mirroring the results for peroxide value reported in Table 1. Furthermore, from a visual pleasantness point of view, the panelists provided higher scores to those samples with a higher degree of ripening.

Finally, Pearson**’**s correlation coefficients were calculated (Appendix A) regarding the VIP discriminant markers of Table 3 and the reported sensory attributes evaluated by panelists. Overall, it was interesting to note that only four compounds established significant correlations (*p* < 0.05) with at least one sensory parameter, namely glutathione (correlated with 6 sensory attributes), proline (correlated only with the “sweet” perception), spermidine (correlated with 3 sensory attributes, namely salty, taste pleasantness and spicy), and γ -glutamyl-glutamic acid (correlated with 6 sensory attributes). Therefore, Pearson’s correlations clearly indicated a hierarchically higher role for peptides in driving the sensory changes observed. In detail, γ -glutamyl-glutamic acid was inversely correlated with global negative aroma (−0.99; *p* < 0.01), though inversely associated with aromatic persistency (−0.99; *p* < 0.01). Another peptide establishing significant correlations was glutathione; this was correlated with odor pleasantness (0.99; *p* < 0.05) and inversely correlated with the intensity of negative odors (0.99; *p* < 0.05), thus suggesting its direct involvement in modulating the sensory profile of Coppa Piacentina during the ripening time. The flavor of glutathione has been widely examined [33] by several sensory evaluations and was found to increase the flavor characteristics of the food matrix, while not affecting the intensity of basic tastes such as sweetness, saltiness, sourness, and umami. Accordingly, we found no significant correlations with sweet and salty attributes, only with those aromatic characteristics of the meat product.

## 3. Materials and Methods

### 3.1. Samples

The different Coppa Piacentina samples were produced and seasoned at a salami factory belonging to the “Salami Piacentini PDO Consortium”, following the procedural guidelines detailed by the European Union regulation on the protection of geographical indications and designation of origins for agricultural products and foodstuffs (EC No. 1263/96). With the aim of studying the evolution of the product during the maturation period, 5 biological replicates of fresh Coppa Piacentina PDO samples were chosen, having undergone 60, 90, 180 and 240 days of ripening. The fresh samples weighed on average 3300 g, while 60, 90, 180, and 240-ripened samples weighed on average 2400 g, 2360 g, 2020 g, and 2000 g, respectively. Samples were taken from the salami factory at the end of August 2022 (under vacuum conditions) and then analyzed for the subsequent panel test (trained panel), chemical analyses, and metabolomics profiling.

### 3.2. Proximate Composition

The chemical analyses were conducted on both fresh (T0) and ripened (T60, T90, T180, T240) Coppa Piacentina samples. In this regard, all the minced samples were analyzed for moisture [34], crude protein (method 984.13; [35]), and ether extract [36] contents. In addition, peroxide number was determined by the AOAC method 965.33 [37] on fat previously extracted from the neck samples in cold conditions according to a modified Foch’s technique [38]. The peroxide number was defined as the amount of peroxide oxygen mEq. O_2_/kg of fat.

### 3.3. Fatty Acids Profile and Total Peroxides

The preparation of the fatty acid methyl esters (FAME) was conducted on approximately 0.2 g of minced Coppa Piacentina samples belonging to different ripening times (i.e., T0, T60, T90, T180, and T240) according to the direct method described by O’Fallon et al. [39]. The FAME were analyzed using a gas chromatograph (model 2025, Shimadzu Corporation, Kyoto, Japan) equipped with an auto-sampler (model AOC-20s, Shimadzu), a flame ionization detector, and a CP-Select CB capillary column for FAME (100 m × 0.25 mm i.d.; 0.25 µm film thickness; Chrompack, Varian, Inc., Palo Alto, CA, USA). The injection volume was 1 mL in split mode (split ratio 30:1) and hydrogen was used as carrier gas at a constant flow of 1.5 mL/min. The injector and detector temperatures were set at 250 °C. The oven temperature program was set as follows: 60 °C for 2 min, from 60 to 170 °C at 10 °C/min for 35 min, and from 170 to 240 °C at 4 °C/min for 9.5 min. The FAME were identified by comparison with the retention times of external standards (Supelco 37 component FAME mix, PUFA-3 menhaden oil, conjugated octadecadienoic acid; Sigma Chemical Co, St. Louis, MO, USA). Data were expressed as a percentage of total fatty acids, calculated with peak areas corrected by factors according to the AOAC 963.22 method [35].

### 3.4. Extraction and UHPLC-HRMS Analysis of Coppa Piacentina PDO Samples

In this work, 1 g of each Coppa Piacentina PDO sample was extracted in 10 mL of a hydro-alcoholic solution (80% methanol, *v*/*v*) acidified with 0.1% formic acid, using a homogenizer-assisted extraction method [4]. The extracts were centrifuged (6000× *g* for 15 min at 4 °C) and then filtered in amber vials (using 0.2 μm cellulose syringe filters) to be analyzed through UHPLC-HRMS.

The untargeted metabolomics analysis was carried out using a Q Exactive™ Focus Hybrid Quadrupole-Orbitrap Mass Spectrometer (Thermo Scientific, Waltham, MA, USA) coupled to a Vanquish ultra-high-pressure liquid chromatography (UHPLC) pump and equipped with a heated electrospray ionization (HESI)-II probe (Thermo Scientific, USA). The mobile phases included water and acetonitrile (both Liquid Chromatography-Mass Spectrometry grade, from Sigma–Aldrich, Milan, Italy), exploiting a gradient elution (6–94% acetonitrile in 35 min) and using 0.1% formic acid as phase modifier. The column used for the separation was ACQUITY UPLC Waters BEH C18 (2.1 × 100 mm, 1.7 µm). The HRMS conditions were previously optimized by our research group [9]. For the full scan analysis, a positive ionization mode with a mass resolution of 70,000 at m/z 200 was used, with a flow rate of 200 μL/min. The injection volume was 6 μL, using a m/z range of 70–1200. Pooled QC samples were randomly injected and acquired in a data-dependent (Top N ions = 3) MS/MS mode with full-scan mass resolution reduced to 17,500 at m/z 200. The fragmentation of the most abundant ions was achieved using collisional energies of 10, 20, 40 eV. The HESI parameters for both MS and MS/MS are reported in Rocchetti et al. [9]. Before the analysis, the mass spectrometer was calibrated using Pierce™ positive ion calibration solution (Thermo Fisher Scientific, San Jose, CA, USA). The collected instrumental data (Thermo.RAW files) were processed using the software MS-DIAL (version 4.70) [40] for automatic peak finding, LOWESS normalization, and annotation via spectral matching, exploiting the comprehensive FooDB database (www.foodb.ca, access date: 15 January 2023). The mass range 100–1200 m/z was searched for features with a minimum peak height of 10,000 cps, using an accurate mass tolerance for peak centroiding of 0.05 and 0.1 Da, for MS and MS/MS, respectively. The identification step was based on mass accuracy, isotopic pattern, and spectral matching. These criteria were used to calculate a total identification score, using as minimum cut-off a value of 50%, considering the most common HESI + adducts. In our experimental conditions, a level 2 confidence in annotation was achieved [41]. Finally, the information regarding the ontology of each annotated compound was provided by the same database used (FooDB).

### 3.5. Sensory Analysis of Coppa Piacentina PDO Samples

The sensory analysis was performed according to the reference ISO 13299 [42] (i.e., *Sensory analysis—Methodology—General guidance for establishing a sensory profile*) and it falls into the category of descriptive–quantitative tests with high informative utility. These tests allow maximal information to be obtained in a short time, using a panel of qualified judges and targeted statistical techniques to achieve results with a high degree of reliability. The Coppa Piacentina PDO samples were all sliced with the same thickness (1 mm). For each sample, two adjacent and overlapping slices were presented, in order to eliminate the transparency factor, thus making the color more saturated. The serving temperature of the product was about 12 °C. The samples tested were presented anonymously, following a precise rotated tasting plan and inserting a reply. Data were processed by The Big Sensory Soft© 2.0 software, using non-parametric statistics.

### 3.6. Statistical Analysis

All the samples of Coppa Piacentina were analyzed in five independent replicates. Data are expressed as the mean ± standard deviation. Statistical analysis of proximate composition, peroxide value and fatty acid profile was performed using SPSS version 21.0 (SPSS Inc., Chicago, IL, USA). The data were subjected to one-way ANOVA with Tukey’s Post Hoc test to determine the differences between samples. Significant differences were compared at a level of *p* < 0.05.

The multivariate data analysis of metabolomics features was performed using three different softwares, namely Mass Profiler Professional B.12.06 (from Agilent Technologies), MetaboAnalyst 5.0 [43], and SIMCA 13 (Umetrics, Malmo, Sweden). Briefly, data were median-centered, Pareto scaled, and Log_2_-transformed before building unsupervised and supervised statistical models, namely PCA and OPLS-DA, respectively. The OPLS-DA models were formerly built considering the impact of ripening time (i.e., at T0, T60, T90, T180, and T240 days). The OPLS-DA model validation parameters (goodness-of-fit R^2^Y and goodness-of-prediction Q^2^Y) were also recorded. Each discriminant model was inspected for outliers, cross-validated by ANOVA (*p* < 0.05), and the permutation testing (number of random permutations = 200) excluded over-fitting. The importance of each compound for discrimination was evaluated using the VIP selection method, using a VIP score threshold of >1. Further, an FC analysis (cut-off > 1.2) was conducted in Mass Profiler Professional B.12.06 to check the variation of each VIP compound for selected comparisons. Finally, Pearson’s correlation coefficients (*p* < 0.05; two-tailed) were calculated considering the VIP discriminant marker compounds and the sensory parameters evaluated by the panelists, using the software PASW Statistics 26.0 (SPSS Inc., Chicago, IL, USA).

## 4. Conclusions

In this work, the combination of several analytical approaches allowed a better understanding of the chemical and sensory changes undergone by a typical Italian PDO meat product, namely Coppa Piacentina. Taken together, the results obtained indicated that ripening time (from 60 up to 240 days) had a marked impact on the potential biomarkers of both oxidative reactions and sensory attributes. The chemical analyses revealed the impact of ripening on the distribution of polyunsaturated fatty acids, which significantly (*p* < 0.05) decreased over time, due to their high susceptibility to oxidation and then conversion to intermediate and secondary molecules. The untargeted metabolomic profiling outlined the involvement of lipid oxidation events in driving the final sensory profile, with some metabolites (such as γ -glutamyl-peptides, hydroperoxy-fatty acids, and glutathione) being particularly discriminant for this purpose. From a visual point of view, the judges involved in the sensory test provided higher scores to long-ripened Coppa Piacentina samples, also outlining potential associations with the lipid biomarkers detected. Taken together, this work confirms the validity of untargeted metabolomics coupled with sensory analysis to evaluate the comprehensive chemical and sensory changes to meat during ripening. Further studies based on the combination of different omics approaches (such as metabolomics and metagenomics) are required to better understand the impact of endogenous and typical Coppa Piacentina PDO microbiota on the chemical modifications observed.

## Figures and Tables

**Figure 1 molecules-28-02223-f001:**
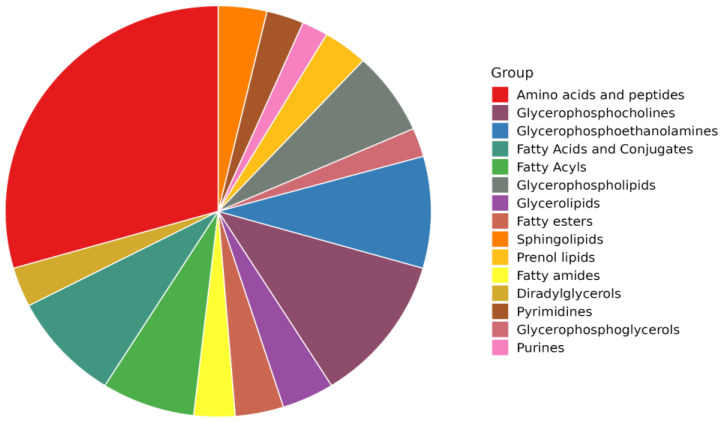
Pie chart diagram showing the distribution of the most abundant classes of compounds in Coppa Piacentina PDO samples, as revealed by untargeted metabolomics.

**Figure 2 molecules-28-02223-f002:**
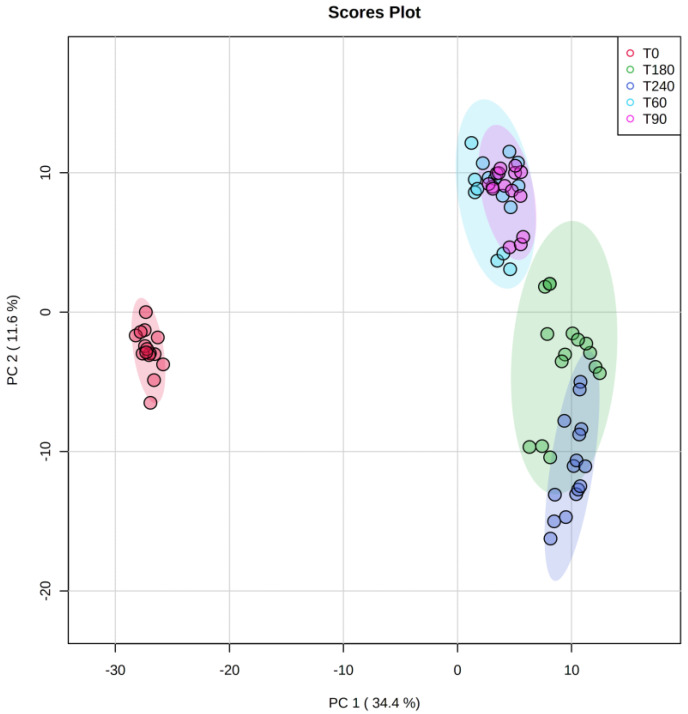
Score plot deriving from unsupervised principal component analysis of the untargeted metabolomic profile of Coppa Piacentina PDO samples during ripening time (i.e., T0, T60, T90, T180, and T240 days). Abbreviations: principal component 1 (PC1); principal component 2 (PC2).

**Figure 3 molecules-28-02223-f003:**
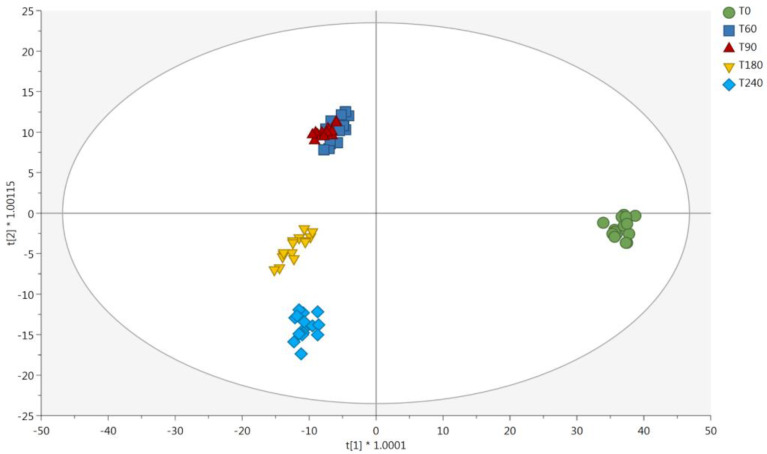
Score plot deriving from supervised orthogonal projection to latent structures discriminant analysis. The output was built considering the untargeted metabolomic profile of Coppa Piacentina PDO samples during ripening time (i.e., T0, T60, T90, T180, and T240 days). Abbreviations: first latent vector (t[1]); second latent vector (t[2]).

**Figure 4 molecules-28-02223-f004:**
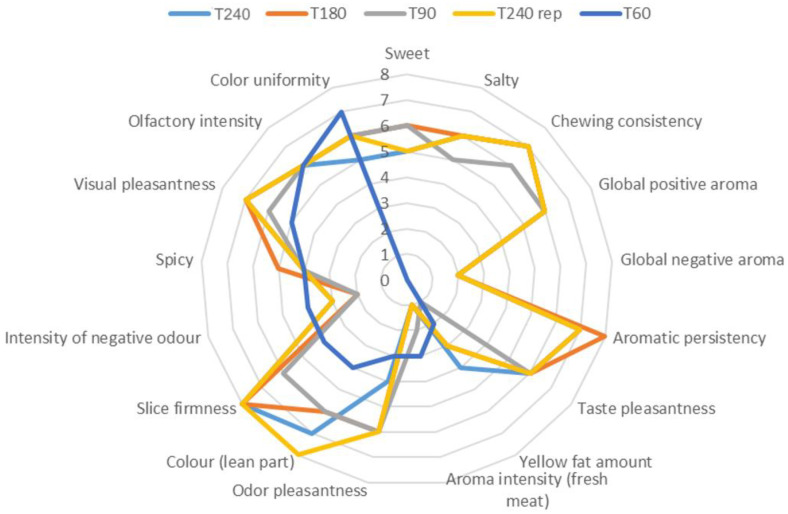
Sensory characteristics of Coppa Piacentina PDO at different ripening times (60, 90, 180 and 240 days). A sample of Coppa Piacentina (i.e., T240 rep) was used before the test as the reference standard to calibrate judges on the different descriptors.

**Table 1 molecules-28-02223-t001:** Effect of ripening time on moisture, protein, lipid and peroxide value of Coppa Piacentina PDO. Results are the mean of five independent analyses.

Ripening Time (Days)	Moisture (g/100 g)	Proteins (g/100 g)	Lipids (g/100 g)	Peroxide Value (mEq. O_2_/kg)
T0	62.3 ± 1.4 ^a^	18.2 ± 0.9 ^a^	17.9 ± 3.3 ^a^	0.6 ± 0.7 ^a^
T60	47.8 ± 2.5 ^b^	20.5 ± 1.3 ^ab^	26.9 ± 3.6 ^b^	10.3 ± 0.8 ^b^
T90	48.8 ±2.1 ^b^	20.9 ± 1.4 ^ab^	25.9 ± 1.1 ^b^	19.8 ± 0.9 ^c^
T180	42.8 ± 3.8 ^b^	21.9 ± 1.6 ^bc^	30.2 ± 4.3 ^bc^	21.2 ± 0.8 ^c^
T240	33.8 ± 4.9 ^c^	24.5 ± 3.3 ^bc^	34.6 ± 4.8 ^c^	26.0 ± 2.4 ^d^

Abbreviations: T0 (fresh Coppa Piacentina; not ripened); T60 (60-days ripening); T90 (90-days ripening); T180 (180-days ripening); T240 (240-days ripening); mEq. = milliequivalents. The superscript letters within each column (^a,b,c,d^) indicate significant differences during the ripening time (*p* < 0.05).

**Table 2 molecules-28-02223-t002:** Fatty acid profile (% total fatty acids) of Coppa Piacentina PDO samples collected at different ripening times. Values followed by different letters within one column are significantly different (*p* < 0.05). nd: not detected (i.e., <limit of detection).

Fatty Acids	T0	T60	T90	T180	T240
C 10:0	0.06 ± 0.01 ^a^	0.07 ± 0.00 ^a^	0.07 ± 0.01 ^a^	0.07 ± 0.00 ^a^	0.07 ± 0.01 ^a^
C 12:0	0.08 ± 0.02 ^ab^	0.09 ± 0.02 ^b^	0.07 ± 0.00 ^a^	0.08 ± 0.00 ^ab^	0.07 ± 0.01 ^a^
C 14:0	1.37 ± 0.01 ^a^	1.33 ± 0.06 ^a^	1.26 ± 0.05 ^a^	1.33 ± 0.10 ^a^	1.32 ± 0.11 ^a^
C 15:0	0.04 ± 0.00 ^a^	0.03 ± 0.01 ^b^	0.03 ± 0.01 ^ab^	0.04 ± 0.00 ^a^	0.03 ± 0.01 ^ab^
C 16:0	27.38 ± 0.08 ^a^	26.57 ± 0.51 ^a^	26.28 ± 0.95 ^a^	26.49 ± 0.74 ^a^	26.63 ± 1.07 ^a^
C 16:1	1.78 ± 0.08 ^a^	1.86 ± 0.20 ^a^	1.73 ± 0.20 ^a^	1.89 ± 0.17 ^a^	2.03 ± 0.17 ^a^
C 17:0	0.19 ± 0.01 ^bc^	0.14 ± 0.03 ^a^	0.18 ± 0.03 ^bc^	0.22 ± 0.01 ^c^	0.17 ± 0.02 ^a^
C 18:0	14.67 ± 0.21 ^a^	14.13 ± 1.08 ^a^	14.59 ± 1.05 ^a^	13.76 ± 0.91 ^a^	14.03 ± 1.15 ^a^
C 18:1n9c	37.20 ± 0.38 ^a^	42.00 ± 1.64 ^bc^	39.18 ± 1.92 ^a^	40.11 ± 1.38 ^abc^	43.10 ± 2.80 ^c^
C 18:1cis11	2.49 ± 0.08 ^a^	2.57 ± 0.15 ^a^	2.41 ± 0.18 ^a^	2.58 ± 0.10 ^a^	2.95 ± 0.19 ^b^
C 18:2n6c	12.13 ± 0.35 ^c^	9.11 ± 1.70 ^ab^	11.89 ± 1.36 ^c^	11.26 ± 1.35 ^bc^	7.58 ± 0.88 ^a^
C 18:3n3	0.52 ± 0.02 ^c^	0.40 ± 0.08 ^b^	0.42 ± 0.03 ^b^	0.42 ± 0.04 ^b^	0.26 ± 0.01 ^a^
C 20:0	0.13 ± 0.01 ^a^	0.14 ± 0.02 ^a^	0.13 ± 0.01 ^a^	0.12 ± 0.01 ^a^	0.14 ± 0.03 ^a^
CLA tot.	0.06	0.03	0.06	0.06	0.05
C 20:1	0.75 ± 0.01 ^ab^	0.67 ± 0.04 ^ab^	0.66 ± 0.09 ^ab^	0.60 ± 0.07 ^a^	0.80 ± 0.15 ^b^
C 20:2	0.50 ± 0.02 ^c^	0.35 ± 0.07 ^ab^	0.45 ± 0.06 ^bc^	0.41 ± 0.06 ^abc^	0.33 ± 0.05 ^a^
C 20:3n6	0.09 ± 0.00 ^c^	0.07 ± 0.02 ^ab^	0.08 ± 0.01 ^bc^	0.09 ± 0.01 ^bc^	0.07 ± 0.00 ^a^
C 20:3n3	0.11 ± 0.00 ^b^	0.08 ± 0.01 ^a^	0.08 ± 0.01 ^a^	0.08 ± 0.01 ^a^	0.07 ± 0.01 ^a^
C 20:4n6	0.38 ± 0.02 ^c^	0.30 ± 0.03 ^ab^	0.36 ± 0.05 ^bc^	0.34 ± 0.04 ^abc^	0.26 ± 0.05 ^a^
C 22:5n3	0.08 ± 0.00 ^b^	0.06 ± 0.01 ^ab^	0.06 ± 0.01 ^a^	0.06 ± 0.01 ^a^	0.05 ± 0.01 ^a^
C 22:6n3	nd ^a^	0.01 ± 0.01 ^b^	0.01 ± 0.00 ^b^	0.02 ± 0.01 ^b^	0.01 ± 0.00 ^ab^
Saturated (SFA)	43.92 ± 0.26 ^a^	42.49 ± 0.94 ^a^	42.62 ± 1.88 ^a^	42.09 ± 1.09 ^a^	42.45 ± 2.21 ^a^
Monounsaturated (MUFA)	42.21 ± 0.51 ^a^	47.10 ± 1.69 ^bc^	43.98 ± 2.23 ^ab^	45.19 ± 1.12 ^ab^	48.88 ± 2.96 ^c^
Polyunsaturated (PUFA)	13.87 ± 0.38 ^c^	10.42 ± 1.91 ^ab^	13.41 ± 1.50 ^c^	12.72 ± 1.40 ^bc^	8.67 ± 0.90 ^a^
Unsaturated (UFA)	56.08 ± 0.26 ^a^	57.51 ± 0.94 ^a^	57.38 ± 1.83 ^a^	57.91 ± 1.09 ^a^	57.55 ± 2.21 ^a^

Abbreviations: T0 (fresh Coppa Piacentina; not ripened); T60 (60-days ripening); T90 (90-days ripening); T180 (180-days ripening); T240 (240-days ripening); CLA = conjugated linoleic acid. Some fatty acids, namely C 11:0, C 13:0, C 14:1, C 15:1, C 17:1, C 18:1t, C 18:2n6t, C 19:0, C 18:3n6, C 21:0, C 22:0, C 22:1n9, C 20:5n3, C 22:2, C 24:0, and C 24:1 have not been identified. The superscript letters within each row (^a,b,c^) indicate significant differences during the ripening time (*p* < 0.05).

**Table 3 molecules-28-02223-t003:** Significant VIP marker compounds related with oxidation processes characterizing the chemical modifications of the different Coppa Piacentina PDO samples during ripening. Each discriminant compound is reported with its VIP score (prediction ability) and Log_2_ FC value against T0.

Discriminant VIP Marker Compound (OPLS-DA)	VIP Score (OPLS-DA)	Log_2_ FC T60 vs. T0	Log_2_ FC T90 vs. T0	Log_2_ FC T180 vs. T0	Log_2_ FC T240 vs. T0
(9S,10E,12Z)-9-hydroperoxy-10,12-octadecadienoate	1.21 ± 0.15	−0.33	−0.10	3.82	5.19
(9Z,11E,13E,15Z)-4-oxo-9,11,13,15-octadecatetraenoic acid	1.16 ± 0.12	3.52	3.65	5.56	6.50
(9Z,11E,14Z)-(13S)-hydroperoxyoctadeca-(9,11,14)-trienoate	1.24 ± 0.15	−0.05	0.77	2.20	3.07
(E)-6-hexadecenoic acid	1.17 ± 0.13	3.69	3.44	2.95	3.45
(Z)-1,4-undecadiene	1.20 ± 0.13	2.45	2.31	4.37	5.30
8-heptadecenal	1.10 ± 0.30	0.44	0.10	1.88	3.16
9-octadecenal	1.12 ± 0.08	2.39	2.49	3.45	3.64
9-tetradecenal	1.18 ± 0.12	1.10	1.64	2.72	3.31
Glutathione	1.12 ± 0.08	−5.97	−6.60	−7.92	−8.14
L-phenylalanine	1.13 ± 0.07	2.22	2.34	2.44	2.58
L-proline	1.12 ± 0.16	1.78	1.49	2.05	1.81
Spermidine	1.10 ± 0.10	1.47	1.90	1.88	1.79
Tyramine	1.18 ± 0.06	1.05	0.82	1.04	1.08
γ -glutamyl-L-putrescine	1.13 ± 0.10	4.97	4.65	6.45	6.83
γ -glutamyl-glutamic acid	1.06 ± 0.38	4.23	3.73	3.77	3.78
γ -L-glutamyl- γ -L-glutamyl-L-methionine	1.11 ± 0.12	7.76	7.34	7.96	7.81

Abbreviations: T0 (fresh Coppa Piacentina; not ripened); T60 (60-days ripening); T90 (90-days ripening); T180 (180-days ripening); T240 (240-days ripening).

## Data Availability

Data are available as Appendix A.

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
