# Peer review of "Untargeted Metabolomics Combined with Sensory Analysis to Evaluate the Chemical Changes in Coppa Piacentina PDO during Different Ripening Times"

_molecules, 2023, doi:10.3390/molecules28052223_

Round 1
Reviewer 1 Report
L 387 – on average
L 392 – what does it mean dry-cured slices..you mentioned before the CP samples considering 60, 90, 180, and 120 days of rippening..the same comment reffers to FA profile analysis
L 449-470 – please describe in detail sensory analysis, e.g. sample preparatin, sampling order, lightening,…
L 450-464 – there is no need to describe in detail descriptive-quantitative test – suggestion..provide some reference to this description or short this part
Conslusion – must be shorter, please conclude, do not repeat results, just conclude…
Tables…please explain in the foother of the table used abbreviations (T0, T60,…)
L 107 – please define the time..what does it mean long ripening period
Table 2 – please put the FA that are not identified in the foother of the table (except C22:6N3 )
L 139 – could you explain why oleic acid rpresented the most abundand FA, and was followed by palmitic, stearic, and linoleic acid.
L 362-364 – how can you explain the degree of rippening with sensory evaluated color parameter
L 370 – what was with rancid smell in the T180 samples
Did the panelists reported some additional remarks for some samples?
L 375-377 – what does it mean visual point of view..overall acceptance or?
Your aim was, among other, to find potential correlations between sensory quality and biomarkers..where is this part calculated and presented by correlations…maybe relationship?
Author Response
Reviewer #1
L 387 – on average
Authors: Added.
L 392 – what does it mean dry-cured slices..you mentioned before the CP samples considering 60, 90, 180, and 120 days of rippening..the same comment reffers to FA profile analysis
Authors: We apologize for the misleading sentence. In this work, five Coppa Piacentina samples for each ripening time (i.e., fresh vs 60, 90, 180, and 240 days) were considered; thereafter, each biological replicate was minced and analysed for both chemical analyses and metabolomics profiling.
L 449-470 – please describe in detail sensory analysis, e.g. sample preparatin, sampling order, lightening,…
Authors: Thank you for the suggestion. The paragraph 3.5 has been revised by adding more details regarding sensory analysis. Please, consider that the sensory test was carried out according to the reference ISO 13299.2.
L 450-464 – there is no need to describe in detail descriptive-quantitative test – suggestion..provide some reference to this description or short this part
Authors: Thank you for the suggestion. The paragraph 3.5 has been revised, accordingly.
Conslusion – must be shorter, please conclude, do not repeat results, just conclude…
Authors: The Conclusion section has been revised, accordingly.
Tables…please explain in the foother of the table used abbreviations (T0, T60,…)
Authors: Done.
L 107 – please define the time..what does it mean long ripening period
Authors: In the paper published by Cava et al. (1999), the authors analysed typical Iberian hams and sampling was done in raw meat, after 210 days of ripening (end of drying) and at the end of ripening (700 days). We have added this information to the revised manuscript.
Table 2 – please put the FA that are not identified in the foother of the table (except C22:6N3 )
Authors: Thank you for the suggestion. The table has been revised, accordingly.
L 139 – could you explain why oleic acid rpresented the most abundand FA, and was followed by palmitic, stearic, and linoleic acid.
Authors: Dear reviewer, the product Coppa Piacentina (owning a European PDO certification mark) is obtained from the cervical muscle of the pig, with a fresh weight no less than 2.5 kg. All the factors included in the PDO certification mark (including feeding system, production in the entire territory of the province of Piacenza, climatic conditions, and technological processing) cumulatively define the quality and chemical composition of the final product; this latter is finally subjected to a ripening time, considering a minimum of six months. In meat science, it is known that beef and pork possess a significantly lower percentage of unsaturated fatty acids (UFA), especially PUFA, and a lower value of PUFA/UFA index than chicken meat. However, in both meat types, the MUFA (monounsaturated fatty acids) share is relatively high and equals 39–48% for beef and 39–52% for pork, and this is strictly dependent from the meat cut considered (https://doi.org/10.3390/molecules26164952). Therefore, the fatty acid composition outlined in Coppa Piacentina PDO analysed in this work is coherent with the meat cut considered (i.e., cervical muscle) and with the overall scientific literature reporting the composition of raw pork neck samples. However, it is always important considering the chemical modification (e.g., lipid oxidation events and further modifications) occurring in the meat matrix during the storage period that can provide exclusive fatty acid profile, such as that observed in this work when considering the high abundance of oleic, palmitic, stearic and linolenic acid and as outlined in a similar available work (https://doi.org/10.3390/molecules26164952).
L 362-364 – how can you explain the degree of rippening with sensory evaluated color parameter
Authors: Thank you for the important question. The 240-days ripened samples were perceived as the most intense in terms of color for the lean part, while samples with a lower ripening degree (60 days) showed a lower color saturation although a more uniform color could be detected. These trends agree with those phenomena occurring during ripening, i.e., a color tending to darken, and a greater surface dehydration causing a more intense color on the outside part, with a corresponding decrease in the color uniformity of the slice.
L 370 – what was with rancid smell in the T180 samples
Authors: With regard to the pleasantness expressed by the judges, the low ripened sample (T60) was not taken into consideration, as it was too young to taste. The sensory results visually revealed that the long ripening period was correlated with a highest pleasantness. However, from an olfactory point of view, the more ripened samples were less appreciated. This trend is not surprising, considering that as the ripening proceeds, the oxidative phenomena intensify, sometimes resulting in the occurrence of rancid odors (usually correlated with a slightly more oxidized outer crown). Regarding the perceived aromas, Coppa Piacentina ripened 240 days was associated with a slight perception rancidness and smoke notes. Taken together, the results of the visual, olfactory and gustatory/tactile evaluation, in terms respectively of more intense colour, oxidized/rancid notes, greater firmness and consistency when chewed, are perfectly in line with what could be expected from the evolution of the product, loosing water during ripening, thus determining a progressive hardness and color saturation. Also, the contact with air for a long time can promote oxidation of the external fat with the development of rancid notes (in domestic consumption it is sufficient to trim any oxidized parts, clearly visible by the consumer).
Did the panelists reported some additional remarks for some samples?
Authors: Regarding some additional remarks outlined by sensory panelists, a reduced perception of other distinctive aromatic notes (such as spices, dried fruit, floral, butter), was observed (although expected). In this regard, it is important to underline that the products analysed are very little spiced (by company practice) and that the rancid and smoke may have covered other more delicate and typifying aromas, probably due to a chemical-physical transformation of lipids and proteins, with the formation of molecules determining a chemical behavior known as unidirectional or bidirectional olfactory suppression/amplification, which could have favored the perception of smoke notes /ash, unexpected to the detriment of other more commonly encountered. Finally, samples ripened 60 days were distinguished by the presence of fresh meat odor, followed by 90 days ripened samples. However, in the more ripened samples, this aroma was not perceived by panelists.
L 375-377 – what does it mean visual point of view..overall acceptance or?
Authors: We referred to hedonic sensory attributes, such as the visual pleasantness perceived by the panelists. The term has been revised, accordingly.
Your aim was, among other, to find potential correlations between sensory quality and biomarkers..where is this part calculated and presented by correlations…maybe relationship?
Authors: Pearson's correlation coefficients have been calculated and discussed in the revised paragraph 2.4. Thank you for pointing it out.
Reviewer 2 Report
Dear Authors, Editors
The manuscript titled “Untargeted metabolomics combined with sensory analysis to evaluate the chemical changes in Coppa Piacentina PDO during different ripening times” needs more improvement before eventual publication. Below are some queries.
v The manuscript format needs to be revised again according to the Molecules template.
v The writing style and language need to be improved a lot. Please check and revise the whole manuscript.
v The authors should follow this structure and rewrite the abstract in brief, including background and problem, the rationale for the study; research objectives; some methodology; important data including statistical analysis; conclusions, novelty, and the importance of the findings.
v Don’t use abbreviations in the title and keywords; write more in-depth keywords.
v The authors should revise the abbreviation in the whole manuscript by following this way: the abbreviations should be defined the first time (and only one time) they appear in each of three sections: the abstract; the main text; the first figure or table (when defined for the first time, the abbreviation should be added in parentheses); then use abbreviation.
v The introduction should be improved with citing the references.
v In the introduction, write two paragraphs more; 1) includes what the ripening of dry-meat method is. As well as advantages and disadvantages of compared with the other methods, 2), about the methods used for the determination of the untargeted metabolomics compared with the applied one in the study as well as write what is metabolomics compounds and their types.
v At the end of the introduction, rewrite the aim of the study clearly with more explanation.
v In the material and methods, the authors should add a section about the chemicals used during this study.
v Please revise the methods part; many sections are in brief; please revise them with a full explanation and mention the applied parameters for the readers, with cite the references.
v Line 416, check the hydroalcoholic solution. Is it only methanol or methanol with acetonitrile?
v Is there any standard for ambient temperature and evaluation time for the whole sensory analysis? Please add this information.
v For Tables 1 and 2, the significant differences should be applied among the treatments. And write the meaning of the letters and the abbreviations in the table footnote.
v In table 2, delete the fatty acids which not detected in all the treatments.
v For table 3, write the meaning of all the treatment abbreviations in the footnote.
v In all the figures, write the meaning of the abbreviation in the figure caption.
v The results and discussion need the authors to revise deeply by comparing them with the previous studies.
v The quality of Figure 4 needs to be improved.
v The references should be revised according to the journal format. No et al. should be in the references.
Author Response
Dear Authors, Editors
The manuscript titled “Untargeted metabolomics combined with sensory analysis to evaluate the chemical changes in Coppa Piacentina PDO during different ripening times” needs more improvement before eventual publication. Below are some queries.
Authors: We would like to thank the reviewer for his feedback and comments about our manuscript. We have carefully revised the paper according to each major drawback.
-The manuscript format needs to be revised again according to the Molecules template.
Authors: We have carefully followed the guidelines of the journal Molecules while preparing the manuscript, using the ad-hoc Manuscript template. In the revised version of the manuscript, we have checked again Figures and Tables format together with the Reference style. Thank you for the revision of the manuscript.
-The writing style and language need to be improved a lot. Please check and revise the whole manuscript.
Authors: Thank you very much for the comment. We have carefully checked and revised both writing style and language. We do believe that the manuscript could be suitable for style and language after this round of revision.
-The authors should follow this structure and rewrite the abstract in brief, including background and problem, the rationale for the study; research objectives; some methodology; important data including statistical analysis; conclusions, novelty, and the importance of the findings.
Authors: The Abstract section has been accurately revised according to reviewer's comments.
-Don’t use abbreviations in the title and keywords; write more in-depth keywords.
Authors: Thank you for the suggestions. The only abbreviation used in the title/abstract section is PDO; this latter is a recognized abbreviation of "Protected Designation of Origin", certification marks widely recognized in EU looking at food products. Also, consider that "Coppa Piacentina PDO" is a direct link with the name and characteristics of this certified product.
-The authors should revise the abbreviation in the whole manuscript by following this way: the abbreviations should be defined the first time (and only one time) they appear in each of three sections: the abstract; the main text; the first figure or table (when defined for the first time, the abbreviation should be added in parentheses); then use abbreviation.
Authors: Thank you for the suggestions. We have deeply checked and revised the abbreviations used.
-The introduction should be improved with citing the references.
Authors: Indeed, the first part of the Introduction refers to the characteristics of the product (i.e., Coppa Piacentina PDO) as detailed by its PDO "Disciplinare di Produzione" also known as "Product specifications"; therefore, no particular references are needed. Overall, few article are available in scientific literature on Coppa Piacentina PDO; therefore, we do believe that the first introductive part on the characteristics of the product is needed. According also to the request below, we have added new references in the second part of Introduction, together with more information on ripening of dry-meat products and untargeted metabolomics approaches.
-In the introduction, write two paragraphs more; 1) includes what the ripening of dry-meat method is. As well as advantages and disadvantages of compared with the other methods, 2), about the methods used for the determination of the untargeted metabolomics compared with the applied one in the study as well as write what is metabolomics compounds and their types.
Authors: added.
-At the end of the introduction, rewrite the aim of the study clearly with more explanation.
Authors: The end of the Introduction has been re-styled and revised in order to strengthen the novelty associated with the aim of the work.
-In the material and methods, the authors should add a section about the chemicals used during this study.
Authors: Thank you for the suggestion. However, considering the wide diversity of chemical analysis and instrumental procedures carried out, we do prefer to cite the chemicals used only in the corresponding paragraphs of Materials and Methods section. We have carefully checked again all the chemicals cited. Thank you for the understanding.
-Please revise the methods part; many sections are in brief; please revise them with a full explanation and mention the applied parameters for the readers, with cite the references.
Authors: Dear Reviewer, regarding the paragraph 3.2. we have cited international and recognized protocols (from AOAC) widely used to analyse the proximate composition of a certain food matrix. We did not report the full description of these methods to avoid plagiarism issues against the M&M part calculated by the iThenticate software of the MDPI platform. Regarding the paragraph 3.3. about the fatty acid profile, we do believe that the method is fully detailed and contain all the information to reproduce the analysis (as also mentioned by another reviewer). Regarding paragraph 3.4. on the metabolomics profiling, we have used a previously published method on a similar matrix (meat), therefore we avoided to give full details that are already reported in a previous work from our research group. However, all the essential information (regarding type of chromatography, MS-equipment, MS analysis and data elaboration) have been clearly provided to the reader. Finally, regarding the sensory test provided in paragraph 3.5., we have deeply revised the method explanation referring to the ISO that was followed to carry out the sensory test together with additional information about the serving of Coppa Piacentina slices.
-Line 416, check the hydroalcoholic solution. Is it only methanol or methanol with acetonitrile?
Authors: the hydroalcoholic solution consisted in methanol 80%; it means, obviously, 80% methanol and 20% water. Therefore, no change is needed this time.
-Is there any standard for ambient temperature and evaluation time for the whole sensory analysis? Please add this information.
Authors: The sensory analysis was performed according to the reference ISO 13299.2 (i.e., sensory analysis-methodology general guidance for the determination of a sensory profile) and it falls into the category of descriptive-quantitative tests with high informative utility. It was performed at ambient temperature and no restrictive evaluation time was given; all the information regarding the sensory test is detailed by the international standard used. We have added this reference into the revised manuscript.
-For Tables 1 and 2, the significant differences should be applied among the treatments. And write the meaning of the letters and the abbreviations in the table footnote.
Authors: The significant differences have been calculated considering the impact of ripening time on each evaluated parameter. The meaning of the letters and the abbreviations have been included as table footnote.
-In table 2, delete the fatty acids which not detected in all the treatments.
Authors: done.
-For table 3, write the meaning of all the treatment abbreviations in the footnote.
Authors: added.
-In all the figures, write the meaning of the abbreviation in the figure caption.
Authors: Thank you for the suggestion. The Abbreviations section has been added below each Table and Figure.
-The results and discussion need the authors to revise deeply by comparing them with the previous studies.
Authors: Indeed, as widely written in the Results and Discussion section, no similar works actually exist dealing with the impact of ripening time on Coppa Piacentina PDO samples. We tried to discuss our data referring to the potential role of each biomarker compounds related to oxidation and to the potential role of meat microbiota. This latter point will be investigated in future works dealing with a combination of metagenomic and metabolomic workflow. Furthermore, in order to meet the reviewer request, we included a discussion in which we compared the fatty acid profile of Coppa Piacentina samples collected during ripening with the ones of other dry-cured pork products Please, just to provide an example, consider that another reviewer of this manuscript reported the following comments: "I really liked the Results and discussion section. It is carefully detailed and very nicely described. The authors worked on this section." We have also added a description of the potential correlations between metabolomics biomarkers and sensory attributes, describing the importance of peptides and dipeptides outlined.
-The quality of Figure 4 needs to be improved.
Authors: The Figure 4 has been provided in .tiff format. We are available to modify the Figure with the Editorial Office before eventual publishing. Thank you for your understanding.
-The references should be revised according to the journal format. No et al. should be in the references.
Authors: The Reference list has been revised, accordingly.
Reviewer 3 Report
The manuscript with the title "Untargeted metabolomics combined with sensory analysis to evaluate the chemical changes in Coppa Piacentina PDO during different ripening times" is a well-written and organized work. It is properly structured and easy to follow.
However, I would have some recommendations for it to be published.
In abstract, I recommend a detailed approach to the purpose of this study.
I really liked the Results and discussion section. It is carefully detailed and very nicely described. The authors worked on this section.
I recommend the statistical analysis in Table 2 and 3. The standard deviation is not enough. With the exception of these tables, the statistics are properly made and detailed.
Reference to Table S1 must be justified
Lines 121-123 detailed
Line 125 how to explain the growth of MUFA
Discussions for fatty acids could be improved
Figure 4 needs to be detailed. Please improve the discussion section regarding the observed results related to this figure.
The Materials and Methods – Samples – section is very little detailed. For such a study, I believe it is necessary to detail a technological flow with the description of raw and auxiliary materials, technological processes and parameters, in order to understand the processes mentioned and analyzed previously.
The description of the analyzes is adequately detailed and can be replicated.
Author Response
Reviewer #3
The manuscript with the title "Untargeted metabolomics combined with sensory analysis to evaluate the chemical changes in Coppa Piacentina PDO during different ripening times" is a well-written and organized work. It is properly structured and easy to follow.
Authors: We would like to thank the reviewer for having appreciated this work.
However, I would have some recommendations for it to be published.
In abstract, I recommend a detailed approach to the purpose of this study.
Authors: We have modified the Abstract section to properly state the aim of the work.
I really liked the Results and discussion section. It is carefully detailed and very nicely described. The authors worked on this section.
Authors: We would like to thank the reviewer for having appreciated the Results and discussion section.
I recommend the statistical analysis in Table 2 and 3. The standard deviation is not enough. With the exception of these tables, the statistics are properly made and detailed.
Authors: We appreciate the reviewer's suggestion. In the previous version of the manuscript, the ANOVA was done in Table 2 only on the principal groups of fatty acids (i.e., SFA, MUFA, PUFA, and UFA). We included the statistical differences on the single fatty acids in the revised version of the manuscript.
Regarding Table 3, it is important to state that each biomarker compound reported derived from a multivariate statistical modelling, namely supervised OPLS-DA and variable importance in projection method (VIP). Accordingly, each biomarker is provided with its discriminant power (VIP score and coefficient of variation) and significant (p < 0.05) Log2 Fold-Change values to highlight the variations against the fresh Coppa Piacentina samples. Therefore, we decided to keep the table 3 in the actual form.
Reference to Table S1 must be justified
Authors: Indeed, considering the omics nature of this work, we usually attach the metabolomics dataset resulting from UHPLC-HRMS analysis as supplementary material, considering that more than 1000 mass features have been putatively annotated. Therefore, to enhance the readability of the manuscript, we use to load all the compounds annotated as supplementary information, thus providing the significant variations only of the discriminant compounds involved in the phenomenon under investigation (Table 3; regarding oxidation events).
Lines 121-123 detailed
Authors: done.
Line 125 how to explain the growth of MUFA
Authors: In Table 2 we have reported the fatty acid profile expressed as relative percentage (g fatty acid/ 100 g of total fatty acids). The % of SFA e UFA remained constant along ripening. However, among UFA, there was a significative change in the distribution of fatty acids between MUFA and PUFA. If the PUFA decrease can be ascribable to their susceptibility to oxidation, the MUFA increase is a mathematical consequence. This result is in agreement with the few works which have already studied the effect of ripening on the lipid profile of other dry-cured pork products (https://www.ncbi.nlm.nih.gov/pmc/articles/PMC4412998/, https://www.sciencedirect.com/science/article/pii/S0308814606002640#aep-section-id27 and https://www.sciencedirect.com/science/article/pii/S0308814602004132). In order to make it clearer, we have modified the sentence in “Moreover, as a direct consequence, a significant increase in MUFA was determined from the raw meat (T0 sample) to the end of ripening (T240 sample), where the proportion ranged from the 42% to the 49% of total fatty acids”.
Discussions for fatty acids could be improved
Authors: The discussion about fatty acid profile has been improved. In particular, we have included a deep investigation in the significative changes of each fatty acid during ripening, as well as additional information about the nutritional properties of PUFA. Finally, we have included a comparison with other works already present in literature which focus on the effect of ripening on the lipid profile of different dry-cured pork products. Unfortunately, it was not possible to make comparisons with more similar works, since the current manuscript represent the first one in which is deeply studied the effect of ripening on Coppa Piacentina PDO biomolecules.
Figure 4 needs to be detailed. Please improve the discussion section regarding the observed results related to this figure.
Authors: the discussion has been improved, accordingly.
The Materials and Methods – Samples – section is very little detailed. For such a study, I believe it is necessary to detail a technological flow with the description of raw and auxiliary materials, technological processes and parameters, in order to understand the processes mentioned and analyzed previously.
Authors: We understand the reviewer's request. However, in this work, we have analysed a typical meat product owning a PDO certification mark. Therefore, the specific details regarding the technological flow, raw materials, and technological processes parameters can be found in the procedural guidelines detailed by the European Union regulation on the protection of geographical indications and designation of origins for agricultural products and foodstuffs (EC No. 1263/96).
The description of the analyzes is adequately detailed and can be replicated.
Authors: thank you.
Round 2
Reviewer 1 Report
No additional comments.
Reviewer 2 Report
Accept
Reviewer 3 Report
The authors responded to all my comments